# Sigma Phase: Nucleation and Growth

**Gláucio Soares da Fonseca \*** , **Priscila Sousa Nilo Mendes and Ana Carolina Martins Silva**

Graduate Program on Metallurgical Engineering, Federal Fluminense University, Avenida dos Trabalhadores, 420, Vila Santa Cecília, 27255-125 Volta Redonda, Rio de Janeiro, Brazil; prisnm22@gmail.com (P.S.N.M.); acmsilva@id.uff.br (A.C.M.S.)

**\*** Correspondence: glaucio@metal.eeimvr.uff.br or glauciosfg11@gmail.com; Tel.: +55-24-2107-3728

**Abstract:** Duplex stainless steels (DSS) and superduplex stainless steels (SDSS) are important classes of stainless steels, because they combine the benefits of austenite and ferrite phases. This results in steels with better mechanical properties and higher corrosion resistance. Owing to these characteristics, DSS and SDSS are widely employed in industry. However, the appearance of undesirable intermetallic phases in their microstructure impairs the properties of DSS and SDSS. Among the undesirable intermetallic phases, the main one is the sigma phase (σ), which can be nucleated when the steel is exposed to the temperature range between 650 °C and 900 °C, reducing the steel's toughness and resistance to corrosion. In a previous work, Fonseca and collaborators used two descriptors of the microstructural path to analyze the formation of sigma phase (σ), the interfacial area per unit volume between sigma phase and austenite ($S_V$), and the mean chord length of sigma (<λ>), both as a function of $V_V$, the volume fraction of sigma, known in the literature as the microstructural partial path (MP). In this work, the contiguity ratio is applied for the first time to describe the microstructural path in the study of sigma phase precipitation in SDSS. The contiguity ratio shows that the distribution of the ferrite/sigma boundaries is homogeneous. Thus, it is reasonable to infer that one has a uniform distribution of sigma phase nuclei within the ferrite. About the kinetics of sigma phase formation, the DSS can be described by the classical Johnson-Mehl, Avrami, and Kolmogorov (JMAK) equation, whereas for the SDSS, the kinetics tend to follow the Cahn model for grain edge nucleation. Finally, we present the three-dimensional (3D) reconstruction of the sigma phase in SDSS. The results demonstrate that the sigma phase nucleates at the edges of the ferrite/austenite interfaces. Moreover, the sigma phase grows and consumes the ferrite, but is not fully interconnected.

**Keywords:** sigma phase; contiguity; kinetics; Cahn models; 3D reconstruction

## 1. Introduction

Duplex stainless steel (DSS) and superduplex stainless steels (SDSS) favorably combine the properties of ferritic and austenitic stainless steels. They possess great mechanical strength, good tenacity, adequate corrosion resistance in various media, and excellent resistance to stress corrosion cracking. These steels are widely used in the chemical industry and in offshore applications that require high corrosion resistance and excellent mechanical strength [1–11]. The continuous development of DSS has resulted in the composition of complex steels containing considerable amounts of alloying elements. The substantial alloying elemental content of these steels aims at obtaining better mechanical properties and superior resistance to corrosion. As usual, the benefits of such additions invariably come with unavoidable drawbacks, the most important being thermodynamic microstructural instability [4,12].

As a consequence of this microstructural instability, during processing (heat treatment, welding) or in use, the precipitation of undesirable intermetallic phases, such as chi (χ), alpha prime (α'), sigma

(σ), carbides, and nitrides can happen. The precipitation of phase σ is of significant interest for DSS and SDSS. The presence of sigma phase worsens the mechanical properties and the corrosion resistance of these steels [1,2,4]. The sigma phase presents the most significant volume when compared to the other intermetallic phases observed in the DSS/SDSS microstructures. Therefore, in many cases, it is common for other phases precipitated in these steels to be disregarded in the analysis of the properties of the material, choosing the sigma phase as the main one responsible for the degradation of such properties [1,2,4–11]. It is of great importance to understand the conditions that lead to the formation of the sigma phase, in order to avoid it during processing or use, and thus preserve the properties of the steel.

These phase transformations, which occur through the nucleation of the new phase and its subsequent growth, are often studied with the support of theories of formal kinetics. The basis of formal kinetic studies is the Johnson-Mehl, Avrami, and Kolmogorov (JMAK) theory. According to JMAK theory, the nuclei are uniform randomly within the matrix. JMAK theory determines the overall rate of phase transformation—that is, the volume fraction as a function of the time of the transformation [13–17]. Using JMAK theory, one can study the kinetics of nucleation and growth of sigma phase. Cahn in his classic work [18] proposed another useful model, which expands JMAK theory. In his work, Cahn models nucleation on the grain boundary (grain face), the grain edges, and the grain corners. In this work, we are going to make use of Cahn's equations extensively. Our study is, to the best of our knowledge, the first study of sigma phase precipitation in SDSS to use Cahn's equations. For example, Sato and Kokawa [19] studied the preferential precipitation site of sigma phase in duplex stainless steel, and concluded that sigma phase formation is strongly affected by the coherence and interfacial energy of ferrite/austenite interface. Recently, Haghdadi and co-authors [20] have studied the tendency of the sigma phase precipitation to take place at the austenite–ferrite interface in duplex stainless steel. They concluded that austenite and ferrite interface with a rational orientation relationship, considering all five crystallographic parameters, which exhibits the lowest tendency for precipitation. Chen and Yang [21] studied the effects of solution treatment and continuous cooling on sigma phase precipitation in a 2205 duplex stainless steel. They suggested that the precipitation of interfacial sigma phase in duplex stainless steel is closely related to the non-coherence of ferrite–austenite and ferrite–$M_{23}C_6$ interfaces. Haghdadi and co-authors [22] have analyzed how two types of austenite morphology, equiaxed and Widmanstätten, influence secondary phase precipitation on exposure to sensitization temperature. They concluded that there is a clear dependence of secondary phase precipitation on the area of the ferrite–austenite interface and the extent of unstable ferrite. These studies [19–22] reinforce the importance of understanding in depth the phenomena of precipitation of the sigma phase in duplex stainless steels.

Using JMAK's equations only, it is difficult to pinpoint the preferential locations of nucleation. The microstructural path (MP) suggested by Gokhale and Dehoff [23] relates the interfacial area density between parent and product phases, $S_V$, and volume fraction transformed, $V_V$. Later, MP was extended by Vandermeer and co-authors [24–28]. In recent work, Fonseca and co-authors [2] used the JMAK theory and the concept of microstructural path with the objective of analyzing the formation, kinetics, and microstructural evolution of the sigma phase in the SDSS. Our previous results indicated that the nucleation of the sigma phase occurs by site saturation with anisotropic linear impingement, i.e., the sigma phase nucleated at the grain edges. However, the microstructure contains further information that can help to characterize the phase transformation. Measuring microstructural quantities in addition to $S_V$ and $V_V$, such as the contiguity ratio $C$ [28], can reveal further microstructural characteristics. The contiguity ratio is the interfacial area between the new phase divided by the total interfacial area. The contiguity ratio ranges from zero to 1—zero in the untransformed condition and 1 in the fully transformed condition. Vandermeer showed that when the contiguity is approximately equal to the volume fraction of the new phase, the nuclei distribution within the matrix is random. A positive deviation from this equality would indicate nucleation in clusters.

The application of the contiguity ratio is frequent in recrystallization studies when new nuclei arise in a deformed matrix. Nonetheless, the ratio has not been employed in the studies of sigma phase precipitation. To use the contiguity ratio in this work, we propose a new definition for it. The new definition is necessary because the microstructures analyzed in this work involve the transformation of the matrix (austenite and ferrite) into sigma phase or sigma phase plus secondary austenite [2]. Thus, in our steels, austenite, ferrite, sigma phase, and secondary austenite may be present.

In contrast, the usual case discussed in the literature involves the decomposition of the parent phase, typically into a new single phase [23]. Recently, Alves and coauthors [29] defined the contiguity ratio for the case of simultaneous and sequential transformations. The authors proposed several equations for the contiguity ratio. Still, in their case, they had to consider interfaces between the three phases. Here, defining the contiguity ratio has an even higher degree of difficulty, because it involves interfaces between four phases.

Therefore, in this work, we intend to analyze the microstructural changes in a DSS treated at 800 °C and an SDSS treated at 750 °C. The temperatures used here are lower than 850 °C. Several authors regard 850 °C to be the temperature at which sigma phase formation [5,9,30] transforms with the fastest kinetics. We studied the formation of the sigma phase at temperatures lower than 850 °C, because we considered these temperatures to be more suitable to study nucleation and growth of sigma phase.

Finally, this work presents a three-dimensional (3D) reconstruction by serial sectioning [31–33] of an SDSS sample treated to reach the thermodynamic equilibrium. Based on this 3D reconstruction, it is possible to analyze the interconnectivity of the sigma phase and how it grows while consuming ferrite.

## 2. Materials and Methods

### 2.1. Duplex Stainless Steel and Superduplex Stainless Steel

Two commercials stainless steels were studied: DSS 2205 and SDSS 2507. The DSS 2205 was hot rolled and annealed, and the SDSS 2507 was hot rolled. Table 1 shows the chemical compositions. Pieces were cut from a delivery plate with 10 mm of thickness to obtain samples with $20 \times 20 \times 10$ mm.

**Table 1.** Chemical composition of duplex stainless steel (DSS) and superduplex stainless steel (SDSS) (wt %).

|       | C     | Si    | Mn   | P     | S      | Cr    | Ni   | Mo   | N     | Cu    |
|-------|-------|-------|------|-------|--------|-------|------|------|-------|-------|
| 2205  | 0.023 | 0.320 | 1.85 | 0.030 | 0.001  | 22.50 | 5.30 | 2.90 | 0.166 | 0.030 |
| 2507  | 0.020 | 0.328 | 0.85 | 0.027 | 0.0009 | 24.89 | 6.82 | 3.72 | 0.278 | 0.156 |

Duplex stainless steel samples were treated at 800 °C for different heat treatment times (up to 79,200 s (22 h)). Superduplex stainless steel samples were treated at 750 °C for different heat treatment times (up to 180,000 s (50 h)) and air-cooled. These two sample groups were used for kinetic studies. Since the kinetics of the sigma phase formation is faster in SDSS [2], the temperature of 750 °C, lower than 800 °C for DSS, was chosen. It was an attempt to study kinetics under the condition of slower sigma phase growth velocity. Another set of SDSS samples was treated at 800 °C for up 108,000 s (30 h). The sample treated for 30 h was chosen for 3D reconstruction.

After the heat treatments (HT), the specimens were ground with emery paper down to 2500 mesh. The samples were metallographic, polished with 6 μm, 3 μm, and 1 μm diamond abrasive. Finally, to reveal the microstructure of the steels, two chemical etchings were carried out. The first etching was for austenitic grain boundaries, the composition of which is 10 mg of picric acid diluted in 10 mL of hydrochloric acid (HCl) for 60 s [34]. The second etching to reveal the sigma phase used a modified Behara solution, made up of 20 mL of hydrochloric acid (HCl), 80 mL of distilled water ($H_2O$), and 0.3 g of potassium metabisulphite ($K_2S_2O_5$); this etching was carried out by immersion during the 1 min and 30 s for sigma phase revelation [1,2].

## 2.2. Stereological Measures

The samples were examined with the Olympus BX51M Optical Microscope (Olympus, Tokyo, Japan) coupled to the Olympus SC30 digital camera. Ten micrographs per sample were captured. The software Stream basic (Olympus, 8.1, Tokyo, Japan) was used to capture the micrographs. After this, micrographs were analyzed with the help of the public domain software Image J, version 1.42q. Volume fractions ($V_V$) and interfacial areas per volume ($S_V$) of ferrite, austenite, and sigma phase were carried out by conventional quantitative metallography techniques [35]. Equations (1) and (2) are as follows:

$$V_V = P_P \tag{1}$$

$$S_V = 2P_L \tag{2}$$

where $P_P$ is the average number of points falling on phase divided by a total number of points applied, and $P_L$ is the average number of intersections per unit length of interfaces with a test (L) line. One can use point counting in Equation (1) to measure the volume fraction of the phases. In each micrograph, with the aid of the software Image J, a grid of 100 points was placed. The sequence of manipulations in the micrographs was: plugins, analyze, grid, lines. This procedure is shown in Figure 4.2, p. 47 of [35]. The interfacial area of the phases is related to "line intercept count", as shown in Equation (2). In each micrograph, five parallel lines were placed on the micrographs with the aid of the software Image J. The sequence of manipulation of the micrographs was: plugins, analyze, grid, horizontal lines. This procedure is shown in Figure 4.7, p. 57 of [35]. In this work, five lines per micrograph were used, with each test line measuring L = 63.6 μm. We prefer to carry out the quantitative analysis manually for each phase, following Equations (1) and (2). Because the samples presented different phases, such as austenite, ferrite, sigma, and secondary austenite. The statistical errors for the volume fraction and interfacial area did not exceed 10%.

Contiguity equations will be deduced in the next chapter.

The volume fraction of ferrite, as well as measured by stereology, was also checked after each time of heat treatment with a Ferritescope (FERITSCOPE® FMP30, Helmut Fischer Gmbh, Sindelfingen, Germany), used to determine the volume fraction of ferrite in DSS. The measurement is non-destructive, fast, and can be performed in situ. The operational principle of the Ferritescope is based on the determination of the magnetic permeability of the material [36].

## 2.3. Computational Thermodynamics

The calculation of the mass quantity of the main elements (Fe, Cr, Ni, and Mo) at 800 °C for the SDSS was performed using the Thermo-Calc software (Thermo-Calc Software, Solna, Sweden) with the TCFE7 database. Also, the mass composition of the previously mentioned elements was measured experimentally, in SDSS samples treated at 800 °C, by energy dispersive spectrometry (EDS), (EDAX, Mahwah, NJ, United States) analysis on a scanning electron microscope (SEM). The Zeiss scanning electron microscope, model EVO MA10 (Carl Zeiss, Stuttgart, Germany), which operates with lanthanum hexaboride (LaB6) filaments. In the sample treated at 800 °C for 108,000 s (30 h), the composition of the sigma phase was closer to that calculated by Thermo-Calc than any other sample. As a result, this sample was chosen for the 3D reconstruction. The idea was to reconstruct the sigma phase as close to thermodynamic equilibrium as possible.

## 2.4. Three-Dimensional Reconstruction

For the 3D reconstruction of the sigma phase, the serial section technique [26–28] was used. This two-dimensional (2D) plane sectioning method comprises several steps, such as proper sample preparation by polishing, etching, microhardness indents, and image acquisition. After a metallographic preparation, like Section 2.1, a field of the sample was randomly chosen to guarantee the continuity of two-dimensional images of the same field, and also to measure the distance between

consecutive sections accurately. For this step, it was necessary to perform Vickers micro-impressions with a load of 980.7 mN for 15 s. The microhardness indents were performed by Shimadzu HMV Vickers Hardness Tester (Shimadzu, Kyoto, Japan). Final polishing took place using a Buehler EcoMet™ 250 Grinder Polisher (Buehler, Lake Bluff, IL, USA). Care should be taken to ensure that the hardness indent is not entirely removed by polishing. If this happened, it would be impossible to align the images and to determine the amount of material removed. The idea was to start with a planar section, where the sigma phase could be detected consuming the ferrite/austenite boundary and proceed with the 3D reconstruction. Our main questions were (1) how is the sigma phase interconnected? (2) Does the sigma phase consume the entire ferrite–austenite boundary? These are questions that need to be answered. For the 3D reconstruction, there was a distance of roughly 0.035 μm between two planar sections. We analyzed about 100 serial sections. Thus, a total of 3.5 μm of depth was removed. In order to reveal the microstructure, the double etching presented in Section 2.1 was performed in each planar section. The reconstructed volume was 127 μm × 68 μm × 3.5 μm. This amount was enough to visualize the sigma phase in 3D and answer the previous questions. Recall that the value of the mean chord length of sigma <λ> at 800 °C is between 1 and 10 μm [2]. With the reconstructed volume, one has much information. To analyze the sigma phase in detail, the sigma phase that consumed the 2D boundary was reconstructed. Only the first 20 sequential images were used. Thus, a small depth of analysis was chosen, so as not to lose microstructural information. To calculate the quantity to be removed ($\Delta h$), the differences between the diagonals ($D_1$–$D_2$) of the microhardness indenter pyramidal was used, with $D_1$ being the diagonal before polishing and $D_2$ the diagonal measured after polishing. Using Equation (3), it is possible to estimate $\Delta h$:

$$\Delta h = \frac{D_1 - D_2}{2\tan\left(\frac{\varphi}{2}\right)} \tag{3}$$

where $\varphi$ is the angle formed between the opposite faces of the pyramid and the square base of the indenter—$\varphi$ is equal to 136°. NIH Image J public domain software, version 1.46r, was used for the three-dimensional processing of images. Two consecutive sections are aligned through common points (such as Vickers micro-impressions in the present study). Sigma phase was reconstructed by cropping and editing images to contain only the sigma phase of interest. The sequence of manipulations in the micrographs was: micrographs transformed into 8-bits; apply enhanced image, contrast, and brightness to differentiate the sigma phase from the other phases; manual alignment using the Align Slice plug-in; correction of the micrographs by making the cut and rendering the volume with the Volume Viewer plug-in. This reconstruction method has been used in many papers with good results [31–33].

## 3. Results and Discussion

### 3.1. Microstructural Characterization

Figure 1 shows an optical micrograph of samples in the as-received (AR) condition for the DSS and the SDSS. In Figure 1a, notice the microstructure with equiaxial austenite grains in a ferritic matrix. This microstructure is a result of industrial annealing after hot rolling. Figure 1b shows the microstructure containing grains that comprise ferrite and austenite lamellae. This lamellar arrangement is commonly found in the literature [1–4,7,8]. The volume fraction of ferrite and austenite for DSS (AR) is 0.55 ± 0.05 and 0.45 ± 0.05, respectively. For SDSS (AR), the volume fraction of ferrite is 0.47 ± 0.03, and for austenite it is 0.53 ± 0.04.

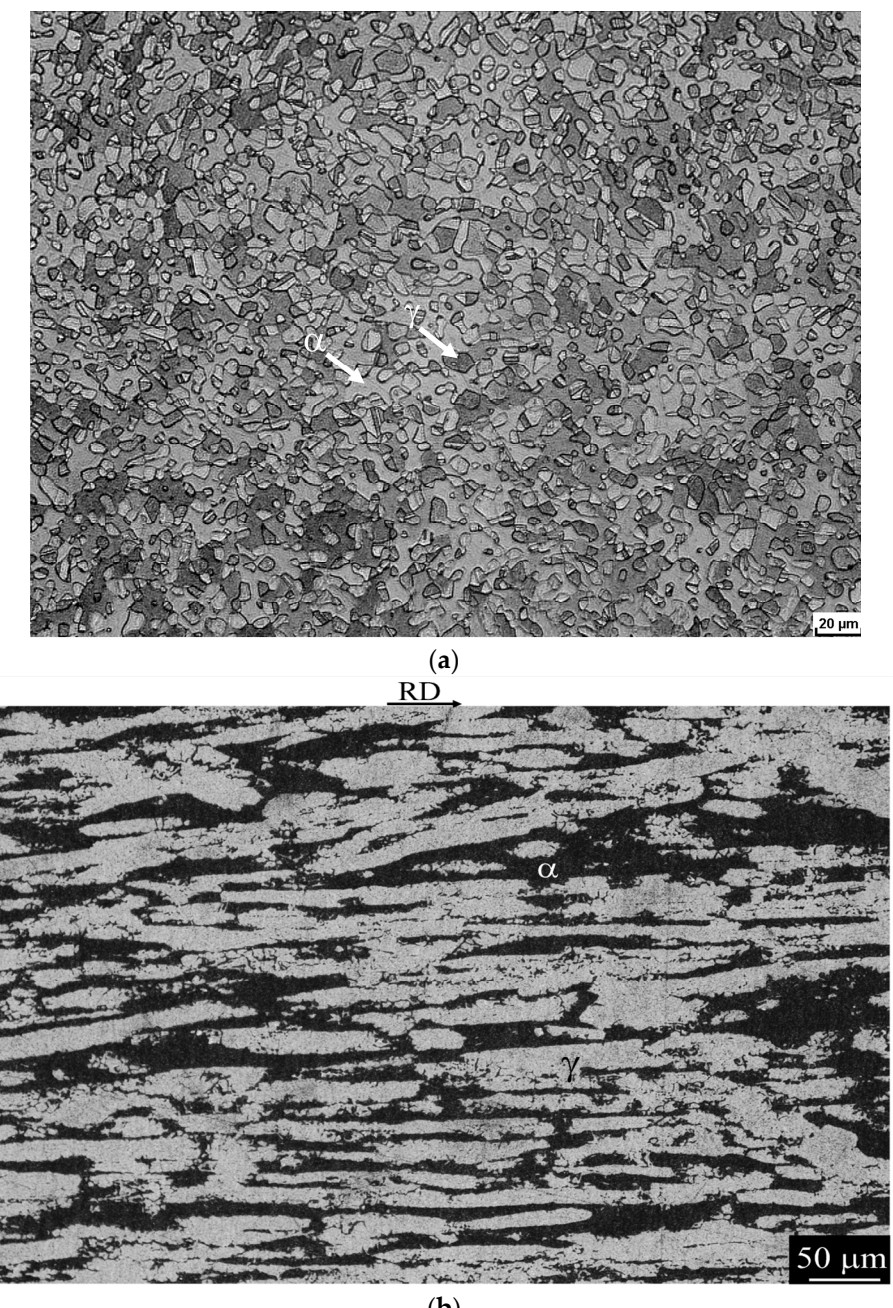

**Figure 1.** (**a**) As-received (AR) micrograph of duplex stainless steel (DSS), showing α-ferrite and γ-austenite. One can see equiaxial austenite grains embedded in the ferritic matrix. (**b**) AR micrograph from superduplex stainless steel (SDSS), showing α-ferrite (black) and γ-austenite (white). One can see elongated grains of both phases. RD: rolling direction.

Figure 2 shows the micrographs of the samples heat-treated at 800 °C for 79,200 s (22 h) for the DSS and 750 °C for 180,000 s (50 h) for the SDSS. Figure 2a displays depleted ferrite, austenite, and sigma phase of DSS, and Figure 2b exhibits austenite, sigma, and secondary austenite of SDSS.

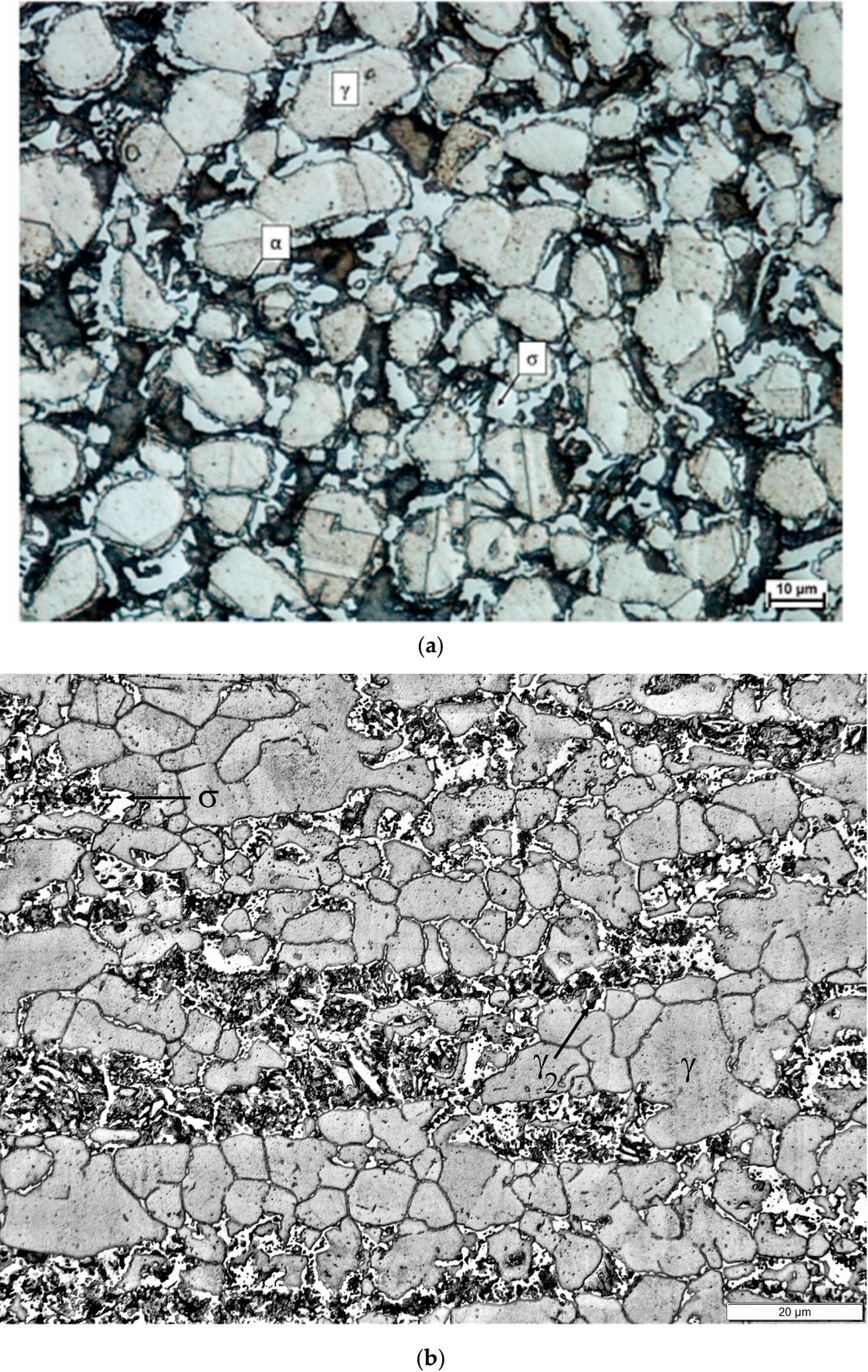

(a)

(b)

**Figure 2.** (**a**) Micrograph of DSS aged 79,200 s (22 h) at 800 °C. One can see grains of austenite (γ), depleted ferrite (α; black), and sigma phase (σ; white). (**b**) Micrograph SDSS aged 180,000 s (50 h) at 750 °C. One can see austenite (γ), secondary austenite (γ$_2$; black), and sigma phase (σ; white).

The volume fraction of ferrite and austenite for the treated samples, shown in Figure 2a, are 0.10 ± 0.01 of ferrite and 0.45 ± 0.01 of austenite in DSS. The sigma phase presented a continuous

morphology. The volume fraction of sigma phase was about 0.45. Thus, ferrite decomposition is given by $\alpha \rightarrow \alpha^* + \sigma$, where $\alpha^*$ is depleted ferrite. The literature of superferritic steels [7] also contains examples of the decomposition of ferrite into depleted ferrite plus sigma phase. Ferritescope measurements confirmed that the volume fraction of ferrite phase remained around 0.10 at the end of the treatment. For the SDSS, the volume fraction of ferrite and austenite for the treated samples, shown in Figure 2b, is around 0.0 for ferrite and $0.55 \pm 0.02$ for austenite. Feritscope measurements confirmed that the ferrite phase was entirely consumed. The eutectoid transformation of ferrite to secondary austenite ($\alpha_2$) plus sigma phase took place in the SDSS. This agrees with several published papers [2,6–8]. The volume fraction of the sigma phase was $0.28 \pm 0.02$, and the volume fraction of the sigma phase plus secondary austenite was $0.48 \pm 0.03$. These values were a result of aging at 750 °C for 180,000 s (50 h).

Duplex Stainless Steel Kinetics

As shown by Magnabosco [4], the kinetics of sigma phase precipitation in DSS in the 700–850 °C range can be represented by the JMAK equation, Equation (4):

$$V_V = 1 - e^{-kt^n} \tag{4}$$

where $k$ and $n$ are constants that depend on the nucleation rate, growth rate, and shape of the precipitates. Magnabosco found $n$ equal to 0.92 for the sample aged at 800 °C. The initial microstructure showed by Magnabosco was a lamellar morphology. In this work, the initial microstructure of the DSS has equiaxed austenite grains (see Figure 1a). One measured the volume fraction of the sigma phase as a function of time aging at 800 °C up to 79,200 s (22 h). Figure 3 depicts the precipitation kinetics of sigma phase. The maximum volume fraction of sigma phase is around 0.45. Figure 3 displays normalized data obtained by dividing the volume fraction of the sigma phase by the maximum volume fraction found at the end of the heat treatment. Equation (4) fitted the data displayed in Figure 3.

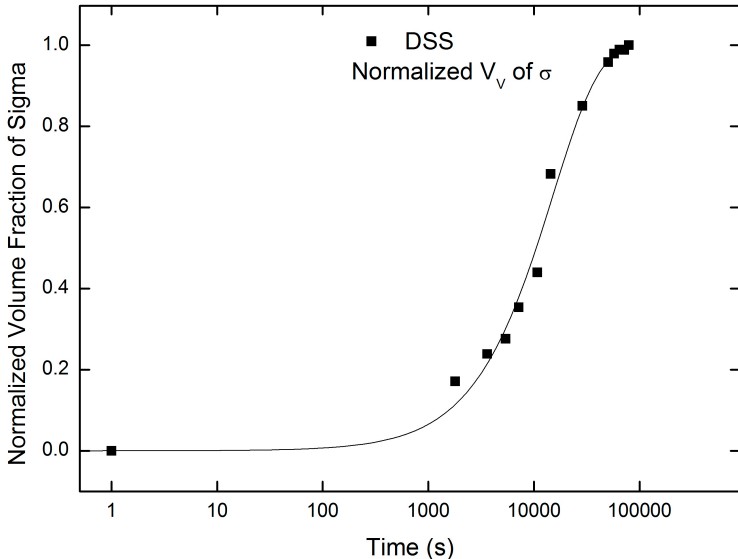

**Figure 3.** Normalized volume fraction of the sigma phase as a function of the aging time, at 800 °C. The solid line represents the fit of the experimental data to Equation (4) (Johnson-Mehl, Avrami, and Kolmogorov (JMAK)).

The fitting parameter obtained using JMAK was $n = 1.0$, with a high coefficient of determination, $R^2 = 0.99$. The value of $n = 1.0$ is very close to 0.92, found by Magnabosco at the same aging temperature. Thus, the initial grain morphology, lamellar [4] or equiaxed, does not appear to influence the kinetics. The value of $n$ near 1 indicates diffusion control, probably of Cr [2,4]. Performing the same procedure

for the SDSS and applying the JMAK model did not yield satisfactory results, as also shown by Magnabosco [4] in treated DSS at 900 °C, and by Fonseca and co-authors in SDSS [2]. Fonseca and co-authors proposed two mechanisms, interface control and diffusion control, to explain the poor agreement of JMAK with the kinetic data.

### 3.2. Superduplex Stainless Steels

Nucleation of the sigma phase needs examination in greater depth. First, would it arise in the ferrite–austenite boundaries? If so, where would nucleation take place: on the grain's face, the grain's edges or the grain's corners? In previous work, Fonseca and co-authors [2] suggested that the sigma phase would nucleate on the grain edges and grow, consuming the ferrite phase. Thus, analyzing the contiguity of the phases becomes paramount to understanding how the sigma phase nucleates and grows. We need to know whether nucleation takes place uniformly or in clusters. Recrystallization studies often employ the contiguity ratio. However, the contiguity ratio can also be useful to understand any solid phase transformation [28]. It would also be of fundamental importance to visualize the sigma phase in 3D and see their interconnections, and how it takes up space in three dimensions. Reconstruction could provide substantial experimental evidence regarding the nucleation site. For readers' clarification, since the main aim was to study the individual interfaces, the contiguity ratio of the ferrite/sigma phase, measured by stereology, plots against the volume fraction of sigma phase. Figure 2b shows that during aging, ferrite decomposes into sigma and secondary austenite, resulting in a typical coral-like morphology [37]. Thus, to interpret the kinetic data, we considered that the amount of ferrite consumed during the transformation was equal to the amount of sigma phase plus secondary austenite formed. Measuring the quantity of $\sigma + \gamma_2$ was possible employing the Feritscope. The Feritscope showed the amount of ferrite after each treatment time. Therefore, the total initial amount of ferrite less the final amount of ferrite was equal to "sigma plus secondary austenite", formed after each treatment time.

Contiguity, Interfacial Migration Rate, and Cahn Models

The contiguity ratio is a stereological parameter, which relates the spatial distribution of new grains—for example, recrystallized in a deformed matrix. One can define the contiguity of the new grains in a deformed matrix, as [28]

$$C_{aa} = \frac{2(S_V)_{aa}}{2(S_V)_{aa} + (S_V)_{ab}} \tag{5}$$

where $(S_V)_{aa}$ is the interfacial area between two new grains, and $(S_V)_{ab}$ is the interfacial area between the new grain and the matrix. Thus, the contiguity ($C_{aa}$) is the fraction of the boundary area of the new grains shared between the new grains. Therefore, the ratio varies from 0, when there is no transformation, to the maximum value of 1, with the complete transformation. Vandermeer showed that when there is a random distribution of new grains, the contiguity value ($C_{aa}$) would be approximately equal to the transformed volume fraction, and that a positive deviation of this equality would signify nuclei clusters. Recently, Alves and co-authors [29] proposed an equation for the contiguity ratio considering a matrix that transforms into two other phases. Therefore, Alves and co-authors extended the concept of contiguity to more than two constituents. Alves and co-authors took into account simultaneous and sequential reactions, obtaining Equation (6):

$$C_1 = \frac{2(S_V)_{11} + (S_V)_{12} + (S_V)_{1M}}{2(S_V)_{11} + 2(S_V)_{22} + 2(S_V)_{12} + (S_V)_{1M} + (S_V)_{2M}} \tag{6}$$

where $(S_V)_{11}$ is the interfacial area between phase 1 and phase 1; $(S_V)_{22}$ is the interfacial area between phase 2 and phase 2; $(S_V)_{12}$ is the interfacial area between phase 1 and phase 2; and $(S_V)_{1M}$ and $(S_V)_{2M}$ are the interfacial areas between phases 1 and 2 with the matrix, respectively. In the present case, it is

necessary to present a new expression for contiguity, since we have a matrix of ferrite and austenite, and part of the matrix transforms into the sigma phase and secondary austenite. It is therefore, possible to have the following interfaces in the microstructure: ferrite–ferrite ($\alpha\alpha$), austenite–austenite ($\gamma\gamma$), ferrite–austenite ($\alpha\gamma$), sigma–sigma ($\sigma\sigma$), ferrite–sigma ($\alpha\sigma$), austenite–sigma ($\gamma\sigma$), secondary austenite–sigma ($\gamma_2\sigma$), and secondary austenite–ferrite ($\gamma_2\alpha$). Numerous combinations of interfaces are possible. In the present study, we assume that the sigma phase nucleates at the ferrite–austenite interface and grows in conjunction with the secondary austenite consuming ferrite. Figure 2b illustrates this point. Equation (7) defines the ferrite/sigma contiguity ($C_{\alpha\sigma}$):

$$C_{\alpha\sigma} = \frac{S'_{V\alpha\sigma}}{2S_{V\alpha\alpha} + S_{V\alpha\gamma} + S'_{V\alpha\sigma}} \tag{7}$$

where $S_{V\alpha\gamma}$ is the interfacial area between ferrite and austenite, $S_{V\alpha\alpha}$ is the interfacial area between ferrite and ferrite, and $S'_{V\alpha\sigma}$ is the interfacial area between ferrite and sigma phase. The values of the interfacial area between secondary austenite–sigma ($S_{V\gamma2\sigma}$) and the interfacial area between secondary austenite–ferrite ($S_{V\gamma2\alpha}$), were added to this ($S_{V\alpha\sigma}$). Thus, we have Equation (8):

$$S'_{V\alpha\sigma} = S_{V\alpha\sigma} + S_{V\gamma2\sigma} + S_{V\gamma2\alpha} \tag{8}$$

During aging, the nucleation and growth of the sigma and secondary austenite phases causes the disappearance of the ferrite–austenite interface, and also of the ferrite–ferrite interface. Figure 4 shows the contiguity of the ferrite–sigma phases obtained by Equation (7) as a function of the volume fraction of sigma. Figure 4 shows SDSS aged at 750 °C. The sigma phase did not consume all other phases present. Consequently, the total volume fraction of sigma was around 0.30 at the end of aging. Thus, it is necessary here to modify Vandermeer's original idea that the contiguity is approximately equal to the volume fraction transformed when random nucleation occurs. Thus, the model presented in Figure 4 considers that $C_{\alpha\sigma} = 3V_{V\sigma}$, where $V_{V\sigma}$ is volume fraction of sigma. The 100% transformation presented for the case of the complete transformation of a deformed matrix to new grains, in the case of the present work, would be around (1/3) transformation.

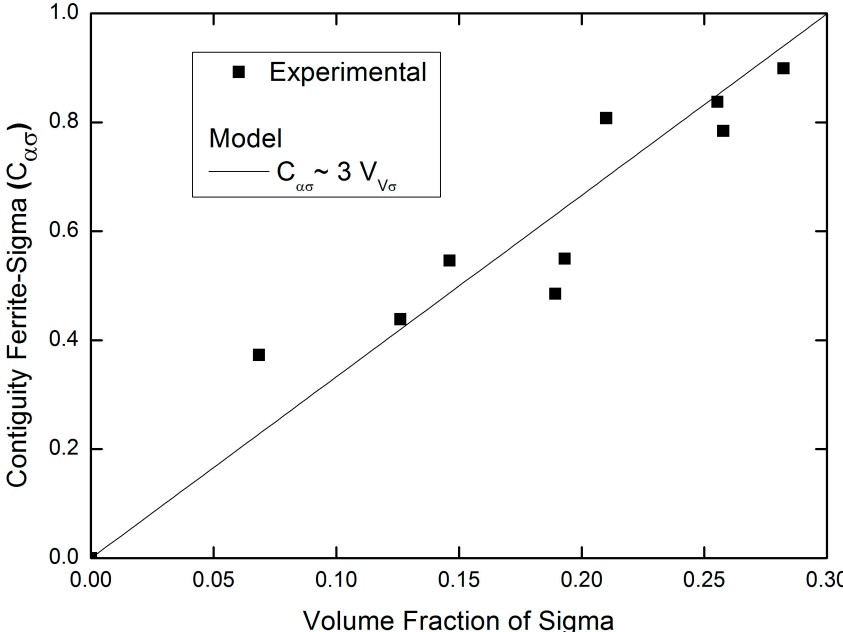

**Figure 4.** Contiguity ratio between the ferrite/sigma phases as a function of the volume fraction of sigma phase. In the case of random nucleation $C_{\alpha\sigma} \cong 3V_{V\sigma}$, the straight line represented by this equation is also presented. $C_{\alpha\sigma}$: ferrite/sigma continuity and $V_{V\sigma}$: volume fraction of sigma.

Figure 4 reveals that the points are close to the model. Therefore, Figure 4 implies a random distribution of the nuclei of the sigma phase within the ferrite. The data do not have a positive linear deviation in relation to the volume fraction of sigma phase. Vandermeer [23] previously studied the austenite decomposition to ferrite in hypoeutectoid steel that exhibits a positive deviation, thus implying that nucleation took place in clusters. In contrast, here we obtained a random distribution of nuclei in space. Such a conclusion, here reported for the first time, was suggested the contiguity between the sigma and ferrite phases in superduplex stainless steel. To calculate the kinetics of formation and growth of the sigma plus secondary austenite, and to apply models different than JMAK—in the case of the present work, the Cahn models [17]—to find the interfacial migration rate (G) of these phases is of fundamental importance. We now derive G, taking into account that the precipitation of sigma plus secondary austenite consumes the ferrite–austenite interface. Figure 5 shows the $S_{V\alpha\gamma}$ variation as a function of time.

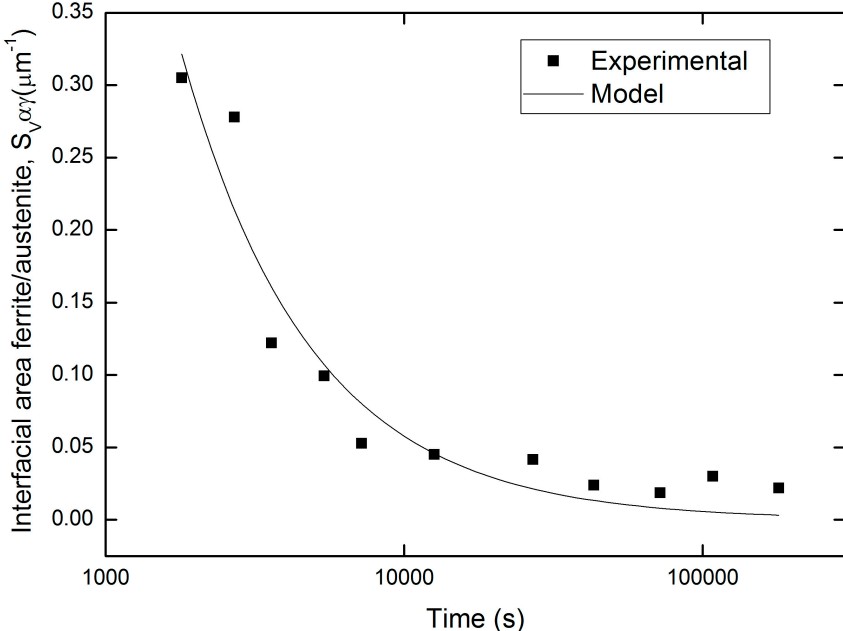

**Figure 5.** Interfacial ferrite–austenite ($S_{V\alpha\gamma}$) area as a function of the aging time, at 750 °C.

Figure 5 demonstrates that during aging, the interfacial area of ferrite with austenite is consumed. Equation (9) corresponds to the model fitted to the experimental data:

$$S_{V\alpha\gamma} = Kt^n \tag{9}$$

where *K* and *n* are constants. Applying Equation (9), we obtain $S_{V\alpha\gamma} = 582t^{-1}$ and $R^2 = 0.92$, a good fit. In addition to analyzing the decrease in the interfacial area ferrite–austenite during the transformation, the volume fraction of ferrite was measured as a function of time, indicating that, as expected, the ferrite volume fraction decreases during aging. Figure 6 shows the volume fraction of ferrite, measured by the Feritscope, as a function of time.

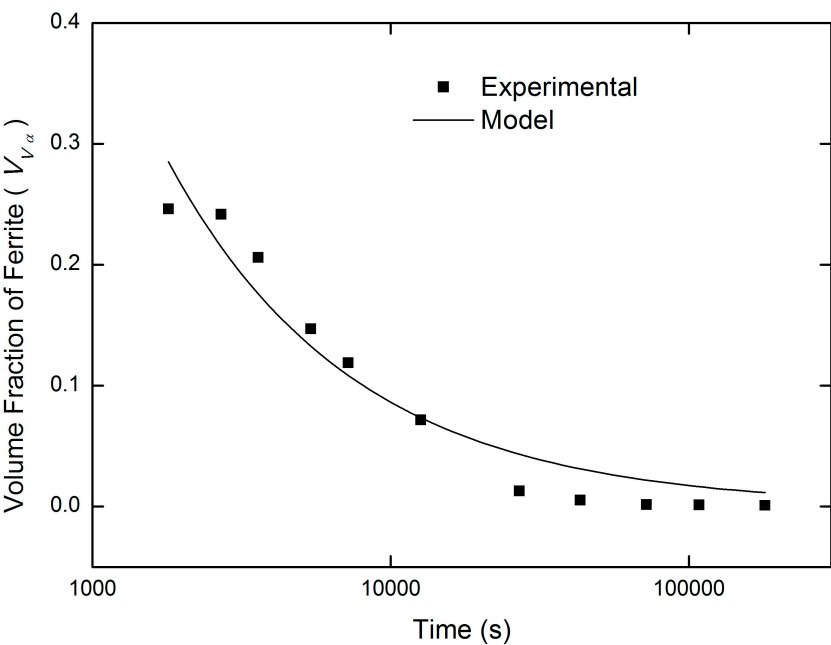

**Figure 6.** The volume fraction of ferrite as a function of the aging time, at 750 °C.

Figure 6 shows that the volume fraction of ferrite as it decreases during aging. The final microstructure at the end of the aging comprises the sigma phase plus secondary austenite (see Figure 2b). The model fitted to the experimental data is presented in Equation (10):

$$V_{V\alpha} = Ct^m \tag{10}$$

where $C$ and $m$ are constants. Applying Equation 10, one obtains $V_V = 52.56t^{-0.7}$ and $R^2 = 0.94$. Therefore, there is good agreement between model and experimental data. With the data obtained from Equations (9) and (10), it is possible to find the interfacial migration rate, $G$. The hypothesis raised here is that the rate of appearance of the sigma plus secondary austenite is equal to the speed of ferrite consumption. The interfacial migration rate $G$ can be estimated by Equation (11), the Cahn and Hagel equation [38]

$$G = \frac{1}{S_{V\alpha\gamma}} \frac{dV_{V_{V\alpha}}}{dt} \tag{11}$$

Combining the results obtained by Equations (9)–(11) gives $G = -0.063t^{-0.7}$. One recognizes here that the growth rate of the sigma plus secondary austenite is equal to the value of $G$ with the opposite sign. Thus, a velocity of roughly 0.00035 µm/s occurs the concentration gradients overlapping the elements, and the velocity decreases [39]. Now, with the knowledge of the interfacial migration rate $G$ of the new phases, sigma plus secondary austenite, it is possible to analyze whether the phases nucleated on the grain boundary, grain edge, or grain corner. For this, we used the classic equations of Cahn [17]:

$$V_V = 1 - e^{-2S_V r} \tag{12}$$

$$V_V = 1 - e^{-\pi L_V r^2} \tag{13}$$

$$V_V = 1 - e^{-\frac{4}{3}\pi C_V r^3} \tag{14}$$

where $S_V$, $L_V$, and $C_V$ are the boundary area, edge length, and grain corner number, respectively, all per unit volume, and $r = Gt$. Equations (12)–(14) are used for the case of grain boundary nucleation,

grain edge nucleation, and corner nucleation, respectively. Since $G$ is not constant, remembering that $G = 0.063t^{-0.7}$, it is necessary to integrate Equation (15):

$$\frac{dr}{dt} = G \tag{15}$$

where we obtain $r = 0.21t^{0.3}$. Knowing that $S_V$, $L_V$, and $C_V$ can be obtained by Equations (16)–(18) [17]:

$$S_V = \frac{3.35}{D} \tag{16}$$

$$L_V = \frac{8.5}{D^2} \tag{17}$$

$$C_V = \frac{12}{D^3} \tag{18}$$

Cahn [17] made these considerations using space-filling tetrakaidecahedral grains. $D$ is the distance between the square faces of a tetrakaidecahedron.

Replacing $r$ with $r = 0.21t^{0.3}$ and inserting the values of $S_V$, $L_V$, and $C_V$ obtained by Equations (16)–(18) into Equations (12)–(14), it is possible to compare the volume fraction of sigma plus secondary austenite obtained experimentally, with the models presented in Equations (12)–(14). Using the initial $S_V$ value of the as-received sample, 0.272 µm$^{-1}$, into Equation (16), one can find $D$. In this case, $D$ is equal to 12.3 µm. With this value of $D$, it is possible to find $L_V$ and $C_V$ using Equations (17) and (18). The maximum value of the volume fraction of sigma plus secondary austenite formed is around 0.48. We normalized the volume fraction of the phases ($\sigma + \gamma_2$) by dividing the volume fraction of sigma plus secondary austenite obtained at each aging time, by the maximum value of $V_V$ found at the end of the treatment. Thus, it was possible to compare the normalized data, shown in Figure 7 with Cahn's models, Equations (12)–(14).

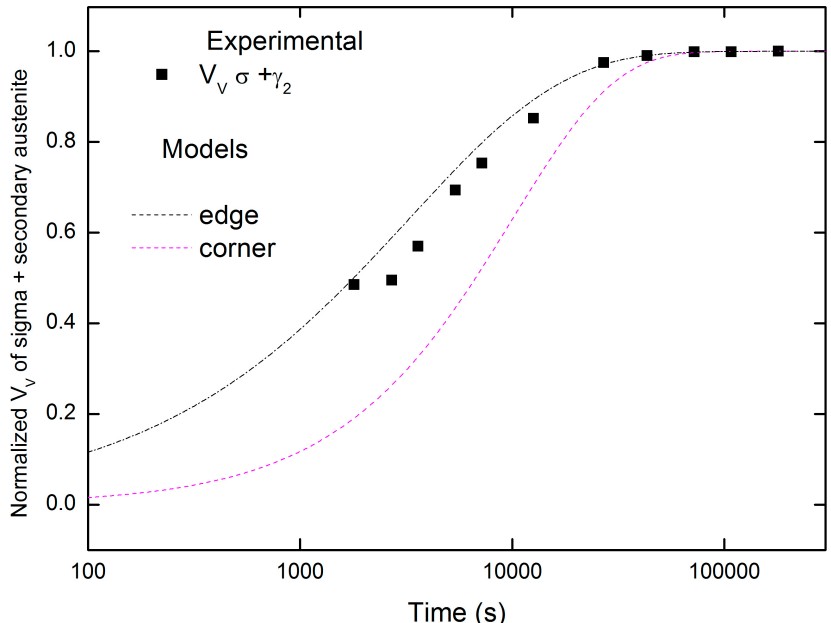

**Figure 7.** The normalized volume fraction of the sigma phase plus secondary austenite as a function of the aging time, at 750 °C. The dashed lines represent the fit of the experimental data to Equations (13) and (14), respectively (Cahn's Model).

Analyzing Figure 7, it is clear the tendency of these phases is to first saturate the edges of the ferrite with austenite. The initial lamellar morphology ($\alpha/\gamma$) seems to have great influence on the

edge length, generating a large number of edges, so that the total edge length was sufficient to provide the nucleation sites for the sigma phase. Haghdadi and co-authors [22] obtained similar results when analyzing the initial morphology of austenite (Widmanstätten or equiaxial) in the precipitation of the sigma phase. They found the greatest extent of the interface area in the DSS with Widmanstätten austenite, compared with the DSS with equiaxial austenite. Figure 7 does not show the grain boundary nucleation model because of the poor fit.

### 3.3. Computational Thermodynamics, Scanning Electron Microscopy, and Energy Dispersive Spectrometry

Thermo-Calc software with database TCFE7 calculated the SDSS equilibrium diagram with the chemical composition presented in Table 1. Figure 8 shows the diagram. Through this diagram, it is possible to extract the mass percentage of the elements at any desired temperature. Table 2 contains the mass percentage of the elements Fe, Cr, Mo, and Ni at 800 °C, in the sigma phase, obtained by the diagram of Figure 8.

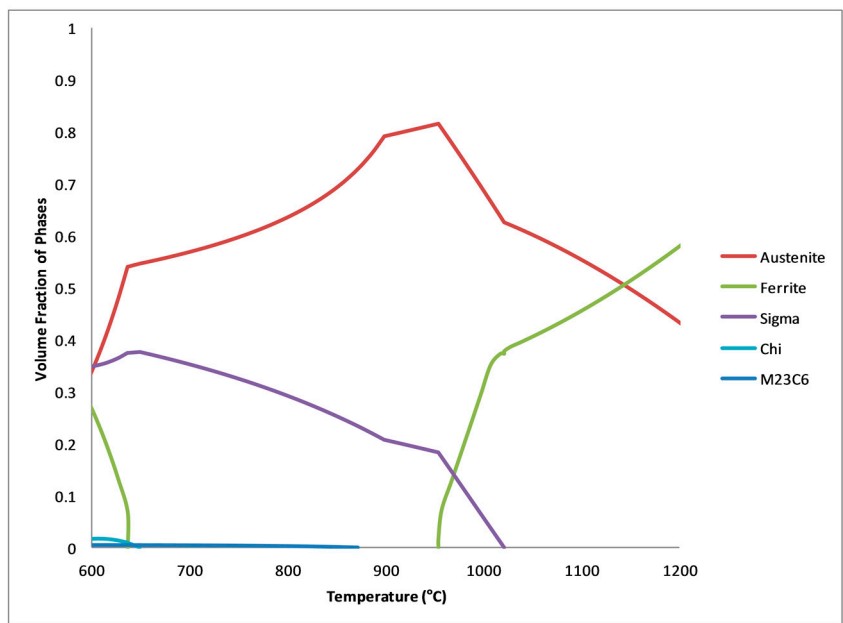

**Figure 8.** SDSS equilibrium phase diagram calculated thermodynamically with the aid of Thermo-Calc software.

**Table 2.** Chemical composition of the sigma phase (mass (%)) at Thermo-Calc 800 °C and experimental 800 °C for 108,000 s.

| Elements | Fe | Cr | Mo | Ni |
|---|---|---|---|---|
| Thermo-Calc | 53.7 | 34.5 | 8.3 | 3.0 |
| Experimental | 53.2 ± 1.0 | 30.2 ± 0.5 | 10.5 ± 0.4 | 6.0 ± 1.0 |

SEM coupled to EDS determined the mass percentage of the sigma phase (Fe, Cr, Mo, and Ni) in the samples treated at 800 °C until 108,000 s (30 h), as shown in Table 2, for comparison with the data provided by Thermo-Calc. Figure 9 shows two micrographs obtained using SEM of the sample treated at 800 °C for 108,000 s (30 h). These micrographs show the locations of the EDS analysis. Each line in the micrograph had an average of 10 reading points. Twenty lines were measured. Thus, a total of 200 points generated the average values presented in Table 2. The sample treated at 800 °C for 108,000 s (30 h) was the one that was closer to the equilibrium values shown in Thermo-Calc.

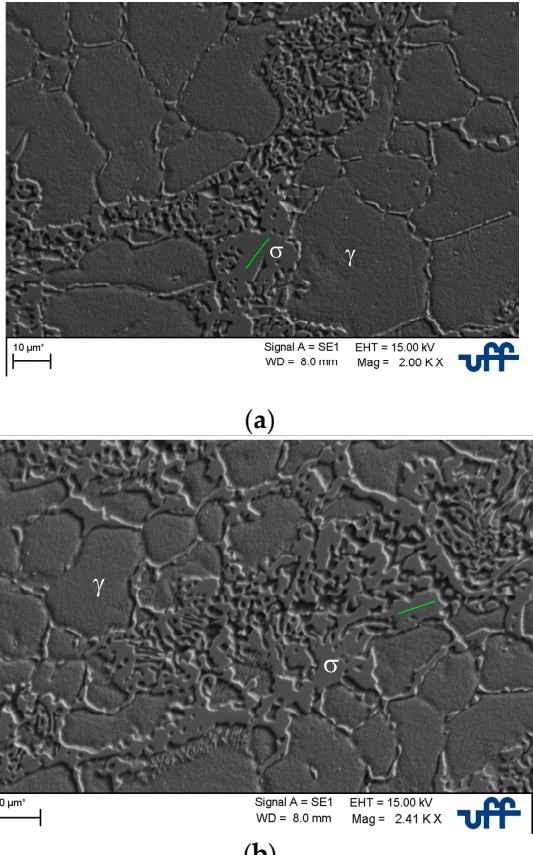

(**a**)

(**b**)

**Figure 9.** (**a**,**b**) Micrographs of the sample treated at 800 °C for 108,000 s (30 h), presenting austenite (γ) and sigma (σ) at different positions of the sample. The green line shows the position of the line of energy dispersive spectrometry (EDS) scanning.

The values found by EDS analysis are in agreement with the composition calculated with the aid of Thermo-Calc software (Table 2). Hence, we chose this sample for the 3D reconstruction, representing the SDSS under equilibrium conditions.

*3.4. Three-Dimensional Reconstruction*

First, one can see the 3D reconstruction of the volume of 127 μm × 68 μm × 3.5 μm of a sample. Figure 10 shows the rendering of the volume. Figure 10 contains too much information. Therefore, it is necessary to specify a smaller region to analyze in detail.

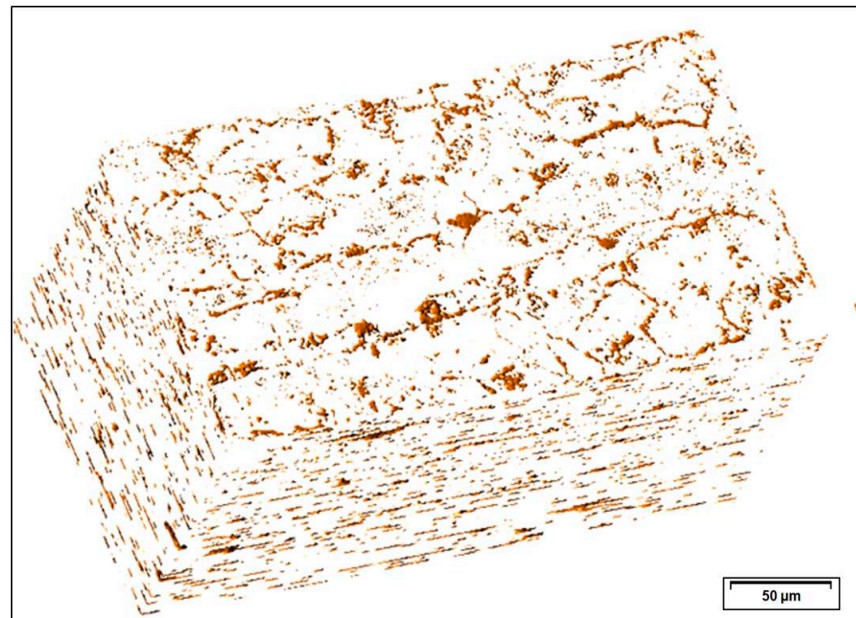

**Figure 10.** Micrograph of the surface-rendered, reconstructed three-dimensional (3D) microstructure. The micrograph depicts the sigma phase in brown.

Figure 11 shows the region chosen for 3D reconstruction. To analyze the sigma phase in detail, we reconstructed the sigma phase that consumed the 2D boundary. We used only the first 20 sequential images. Despite a reduced number of layers, the 3D analysis was sufficient to indicate the preferred nucleation site.

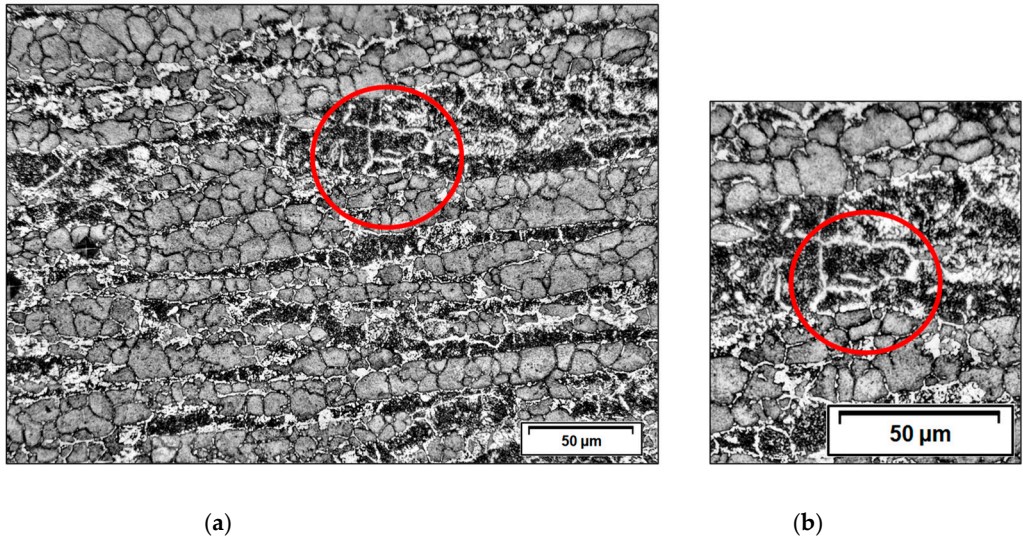

(**a**)                                                                                    (**b**)

**Figure 11.** (**a**) Micrograph of the sample treated at 800 °C for 108,000 s (30 h), presenting the region of interest for reconstruction within the red circle; (**b**) the boundary that will be reconstructed, shown in detail.

Figure 12 shows the region analyzed from several angles. The sigma phase consumes the edges of the ferrite–austenite interface, and grows into the ferrite boundary. However, the boundary is not fully consumed, but the edge is almost entirely consumed. This finding agrees with Figure 7—that is, the sigma phase first saturates the edges of the ferrite with austenite. The sigma "wets" the edge, but it is absent in some points, interrupting the interconnection of the boundary. This is similar to Kral and coauthors' [27] reconstruction of proeutectoid ferrite. Here, we present the first experimental evidence

that the sigma phase nucleates at the edges of ferrite–austenite. The 3D reconstruction corroborates the findings from the kinetic data (see Figure 7). Therefore, the grain boundaries, i.e., the grain faces, are not preferred sites for sigma phase nucleation. The grain edges are the preferred sites for sigma phase nucleation.

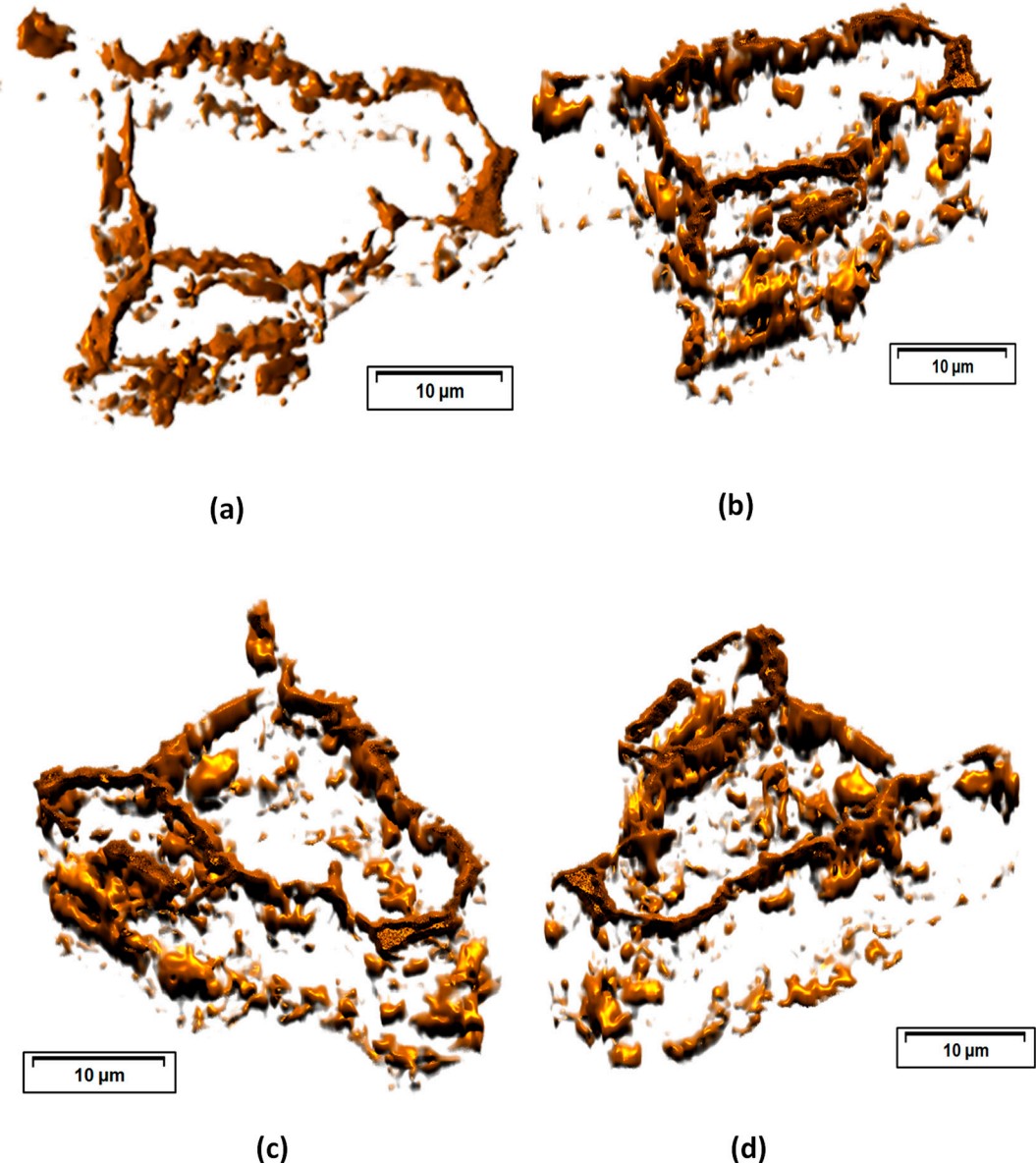

(a)　　　　　　　　　　　　　　　　　　　　　　　　　(b)

(c)　　　　　　　　　　　　　　　　　　　　　　　　　(d)

**Figure 12.** Sigma phase reconstructed in 3D. (**a**–**d**) Different viewing angles of the reconstructed region, shown in Figure 11.

## 4. Conclusions

The main conclusions of the investigation of precipitation kinetics of the sigma phase in duplex stainless steel and superduplex stainless steel were:

- Classical JMAK equations can adjust the kinetics of sigma phase precipitation in DSS with equiaxial morphology of the austenite grains. Ferrite decomposition during aging generated depleted ferrite plus sigma, as found in superferritic stainless steels. Ferrite decomposition during aging in SDSS generated, as expected, sigma phase plus secondary austenite.

- For the first time, the microstructure of SDSS was studied, with the help of the contiguity ratio of ferrite/sigma. According to the contiguity ratio, the distribution of the nuclei of the sigma phase is random, and $C_{\alpha\sigma} = 3\,V_{V\sigma}$.
- Cahn's equation for grain edge nucleation fitted well with the kinetics of precipitation of the sigma phase plus secondary austenite in SDSS.
- Computational thermodynamics indicated that the composition of the sigma phase aged at 800 °C for 108,000 s (30 h) was close to equilibrium.
- The 3D reconstruction, using the sample with the composition closest to equilibrium, verified that the sigma phase occupies the edges of the ferrite–austenite interfaces, and grows around the austenite. The sigma phase loses the interconnection at some points on the grain boundary. For the first time, there is experimental evidence of sigma phase nucleation at the ferrite–austenite edges. In other words, the grain edges are preferred sites for the nucleation of the sigma phase.

**Author Contributions:** G.S.d.F. conceived and designed the experiments; P.S.N.M. performed the SDSS experiments; A.C.M.S. performed the DSS experiments and computational thermodynamics; G.S.d.F. analyzed the data and wrote the paper.

**Funding:** This research was funded by Fundação de Amparo a Pesquisado Estado do Rio de Janeiro (FAPERJ), grant number: E-26/203.238/2016; Conselho Nacional de Desenvolvimento Científico e Tecnológico (CNPq); and Coordenação de Aperfeiçoamento de Pessoal de Nível Superior (CAPES)—Finance Code 001.

**Acknowledgments:** Thanks go to Paulo R. Rios for helpful discussions and critical reading of the manuscript.

**Conflicts of Interest:** The authors declare no conflict of interest. The funders had no role in the design of the study; in the collection, analyses, or interpretation of data; in the writing of the manuscript, or in the decision to publish the results.

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
