# Peer review of "Sigma Phase: Nucleation and Growth"

_metals, doi:10.3390/met9010034_

Reviewer 1 Report

The authors have studied the nucleation and growth of Sigma in duplex stainless steels. The contiguity ratio has been applied to describe the sigma precipitation in SDSS. A difference in the Sigma phase formation kinetics has been reported between DSS an SDSS. The work is interesting and worth publishing. The authors are however suggested to consider the following works on duplex stainless steel in their introduction and discussion:

1- Scripta Mater. 40 (1999) 659–663  (interface orientation dependence of Sigma precipitation)

2- Materials Letters 196 (2017) 264–268 (interfacial plane dependence of Sigma precipitation)

3- Materials Science and Engineering: A, 311(2001) 28-41 (solution treatment and continous cooling dependence of Sigma precipitation)

4- Materials Letters 238 (2019) 26-30 (Austenite morphology dependence of Sigma precipitation)

5- Corrosion Science, 51(2009) 1850-1860  (effect of phase boundary)

Author Response

The authors would like to thank the reviewers for their valuable comments and careful analysis of our paper. We carefully addressed all the reviewers’ suggestions. The text changes are highlighted in yellow.

The authors have studied the nucleation and growth of Sigma in duplex stainless steels. The contiguity ratio has been applied to describe the sigma precipitation in SDSS. A difference in the Sigma phase formation kinetics has been reported between DSS an SDSS. The work is interesting and worth publishing. The authors are however suggested to consider the following works on duplex stainless steel in their introduction and discussion:

1- Scripta Mater. 40 (1999) 659–663  (interface orientation dependence of Sigma precipitation)

2- Materials Letters 196 (2017) 264–268 (interfacial plane dependence of Sigma precipitation)

3- Materials Science and Engineering: A, 311(2001) 28-41 (solution treatment and continuous cooling dependence of Sigma precipitation)

4- Materials Letters 238 (2019) 26-30 (Austenite morphology dependence of Sigma precipitation)

5- Corrosion Science, 51(2009) 1850-1860  (effect of phase boundary)

R: All suggested articles were used to improve the introduction. Moreover,  the most recent article was also used to improve the discussion. We revised the English as best as we could.

Reviewer 2 Report

Dear Authors,

 I would like to suggest youconsidering some general modification in your paper:

-more detailed desciption  of the experimental  procedures: microstructure revealing, observation and its image quantitative analysis, taking into account possible artefacts and introduced  uncertainty field, as it is crucial for  further  consideration of  models JMK, Cahn etc.

- modification of  Figures 3 a and  3b, 8a and 8b.  as they  contain practically  the same informations, and  Figures 4 and 5 as they could be compiled with Fig3.

 Some detailed remarks are in attached file.

Author Response

The authors would like to thank the reviewers for their valuable comments and careful analysis of our paper. We carefully addressed all the reviewers’ suggestions. The text changes are highlighted in yellow.

Reviewer 2

Dear Authors,

 I would like to suggest you considering some general modification in your paper:

-more detailed description  of the experimental  procedures: microstructure revealing, observation and its image quantitative analysis, taking into account possible artefacts and introduced  uncertainty field, as it is crucial for  further  consideration of  models JMK, Cahn etc.

R: These considerations were answered in the reviewer's questions presented in the manuscript. The questions are transcribed below

- modification of  Figures 3 a and  3b, 8a and 8b.  as they  contain practically  the same informations, and  Figures 4 and 5 as they could be compiled with Fig3.

R: We agree with the reviewer. Therefore, we removed Figures 3a and 8a from the article. I believe that Figures 4 and 5, should be presented as it is in the paper. Thus, it becomes clearer for the reader, in our opinion.

 Some detailed remarks are in attached file.

We give our answers to the remarks contained in the reviewer’s attached file below:

1)      what was the  general  difference between heat treatment and isothemal aging?

R: We decided to delete the expression “isothermal aging” as it does not reflect the variety of situations that might happen during the steel usage.

2)      How do you define grain boundary and grain edge?

R: That is a good question. “grain boundary” is normally used to mean a grain face as we explain in what follows. A polyhedral grain comprises grain faces, grain edges (where two grain faces meet) and grain corners (where three grain faces meet). Some authors even distinguish grain face from grain boundary; grain edge from triple junction and grain corner from quadruple junction. These authors consider that faces, edges, and corners apply to the polyhedron itself whereas boundary, triple junctions and quadruple junctions apply to polycrystalline networks. In this work, we use grain boundary to mean grain face and grain edges and corners. Because of the reviewers comment we inserted “grain face” in parenthesis after “grain boundary” in the first time it appears to clarify this point. We adopted this nomenclature after J. W. Cahn who uses: grain boundary, grain edge and grain corner in his papers.

3)      Some  more details should be added: digital or analog microscope  images were examined, how were  detection thresholds established for particular  phase and for both Vv and Sv parametr whether  the used software did not give possibility to estimate directly   grain perimeter and this  simplified   chords meausrement method was used

R: We inserted text explaining the experimental procedure in more detail. See lines 145-165

4)      What about grains boundaries revealing after each step of serial sectionning:: either they were  visible  in a satisfactory way or an additional  etching  had   to be carried out each time  Thererefore, how the the repeatabilty of the  stereological  analysis for each cross-section were  established .  

R: The specimen had to be etched after the polishing removed it. We inserted the etching procedure in the text. Lines 200-201. We employed standard stereological methods. We believe that this methodology has proven to be adequate in many previous papers.

5)      The same questions as above.

R: As we answered above, this reconstruction method has been used with reliable results in many papers. We cite the more relevant papers in the text refs.31 a 33.

6)      Results of the 3D reconstruction  show that  signa phase particles are not always convex, therefore each analysed point could  contain  an information  not only from sigma phase partices but from matrix also ( 3D detection volume) thus microanalysisi results obtained for several subseqeunt  cross-section would be more adequate than those for many particles situated on  the same cross section.

R: The reviewer is in principle correct. Of course, the analysis would contain information not from the sigma phase if the sigma phase regions were smaller than the volume sampled by the EDS analysis. However, in our case, the dimensions of the sigma phase were larger than the volume sampled by the EDS analysis. Therefore, we believe that the analysis is consistent. Furthermore, our measurements are consistent with Thermo-Calc calculations.

7) Dispersion degree and surface  development of sigma phase particles visible in fig.12 are very high, thus the question is: how and if  the procedure of manual establishing of the detection threshold for image analysis   was evaluated.

R: As described in the text, the following procedure was used: “The sequence of  manipulations in the micrographs was: micrographs transformed into 8-bits; apply enhanced image, contrast, and brightness to differentiate the sigma phase from the other phases; manual alignment  using the Align Slice plug-in; correction of the micrographs by making the cut and rendering the  volume with the Volume viewer plug-in”. This procedure is standard for image analysis with Image J software. Therefore, we believe we used the best and widely accepted experimental procedures.

Reviewer 3 Report

In the manuscript, authors give an overview of sigma phase evolution in stainless steel. Manuscript is well written and clear. I suggest to improve introduction to give more clear overview why sigma phase should be investigated in such many details. There are several technical suggestions:

I suggest to change title, while it is too general for large research manuscript. More selfexplaining title would help to attract more general readers.

Figures 5, 6, 7 should be probably combined in one single plot, while fig 7 gives absolutely no information as well as its capture should be changed to be more clear.

ables 3 and 4 can be probably given as text, while both are too small or should be combined in one single table.

Author Response

The authors would like to thank the reviewers for their valuable comments and careful analysis of our paper. We carefully addressed all the reviewers’ suggestions. The text changes are highlighted in yellow.

In the manuscript, authors give an overview of sigma phase evolution in stainless steel. Manuscript is well written and clear. I suggest to improve introduction to give more clear overview why sigma phase should be investigated in such many details. There are several technical suggestions:

R: We revised the introduction and included more references.

I suggest to change title, while it is too general for large research manuscript. More self-explaining title would help to attract more general readers.

R: We appreciate the reviewers’ suggestion. However, after thinking about it, we concluded that the current title is quite good. So we would like to keep it.

Figures 5, 6, 7 should be probably combined in one single plot, while fig 7 gives absolutely no information as well as its capture should be changed to be more clear.

R: We agree that fig. 7 could be removed. However, We believe that Figures 5 and 6 should be left as they are. It is highly unusual to combine two different parameters such as volume fraction and interfacial area in the same plot.

Tables 3 and 4 can be probably given as text, while both are too small or should be combined in one single table.

R: They were combined, as requested.

Round  2

Reviewer 1 Report

I think the revisions are satisfactory.